# Season and Nitrogen Fertilization Effects on Yield and Physicochemical Attributes of Strawberry under Subtropical Climate Conditions

**Shinsuke Agehara** [1,*] and **Maria Cecilia do Nascimento Nunes** [2]

[1] Gulf Coast Research and Education Center, Institute of Food and Agricultural Sciences, University of Florida, 14625 CR 672, Wimauma, FL 33598, USA

[2] Food Quality Laboratory, Department of Cell Biology, University of South Florida, Tampa, FL 33620, USA; mariacecilia@usf.edu

\* Correspondence: sagehara@ufl.edu; Tel.: +1-813-419-6583

**Abstract:** Strawberry (*Fragaria ×ananassa* Duch.) yields in winter production regions are greatly affected by early-season nitrogen (N) fertilization, especially when pre-plant N is not applied. In Florida, United States, applying N at high rates during the early season is a common fertilization practice, but little is known about its impact on fruit quality. The objective of this study was to examine season and early-season N fertilization effects on yield and physicochemical attributes of 'Florida Radiance' strawberry grown under subtropical climate conditions. Field experiments were conducted in west-central Florida over two growing seasons. Plants were treated with three N rates (1.12, 1.68, and 2.24 kg/ha/d) over 21 days during the early vegetative growth stage. Thereafter, all plants were treated with the same N rate of 1.12 kg/ha/d until the end of the season. Increasing the early-season N rate increased marketable yield by 15% to 18%, but it had no significant effect on any fruit quality attributes. Contrarily, marketable yield was similar in both seasons, whereas fruit quality showed remarkable seasonal variations. In the season with higher solar radiation and lower temperature, RH, and rainfall during the fruit development period, berries were redder with increased anthocyanin accumulation but had lower pH, acidity, and soluble solids. These results suggest that season has a greater effect on fruit quality than early-season N fertilization, which is complex to dissect because of the interaction between fruit quality attributes and environmental conditions. The use of high N rates during the early season appears to be an effective strategy to improve the profitability of winter strawberry production. Importantly, this fertilization technique has a minimal risk of compromising fruit quality or fertilizer N use efficiency.

**Keywords:** anthocyanin; fertility; fertilizer; Florida; *Fragaria ×ananassa*; fruit quality; nutrient management

## 1. Introduction

Strawberries (*Fragaria ×ananassa* Duch.) are one of the most popular and consumed berries worldwide because of their attractive shape, color, and flavor. Furthermore, strawberries are an excellent source of bioactive compounds such as vitamin C and polyphenols [1]. However, strawberry fruit quality can vary from season to season because environmental conditions affect physical, chemical, and sensory characteristics [2,3]. Fruit quality is also affected by several pre-harvest factors, including harvest time, fruit maturity, diseases, and fertilization [3–10].

Seasonal variations in strawberry sensory and physicochemical profiles have been reported in diverse strawberry producing regions [5,7,8,11]. In Spain and North Carolina, United States, for example, it has been reported that strawberries had lower acidity but higher soluble solids content (SSC), ascorbic acid, total phenolics, and total anthocyanins when harvested later in the season [7,11]. Agüero et al. [5] reported that air temperature

did not affect the SSC of several strawberry cultivars grown in Argentina, whereas acidity increased as temperature raised during the season. In Florida, United States, Cayo et al. [8] found reductions in SSC and anthocyanin accumulation in several strawberry cultivars harvested later in the season when field temperatures were higher.

Strawberries have a long harvest window, requiring frequent and intensive fertilization. Among mineral nutrients, nitrogen (N) is generally the most limiting nutrient for crop production [12]. In winter strawberry production, early-season N fertilization greatly affects fruit earliness and yields, especially when pre-plant N is not applied [13]. Florida is the largest winter strawberry producing state in the Unites States. In west-central Florida, our previous study [13] tested five early-season N rates (0.22, 0.67, 1.12, 1.57, and 2.02 kg/ha/d) for two major strawberry cultivars. Increasing the early-season N rate increased early- and total-season yields by up to 65% and 58%, respectively. Model fitting analysis revealed that yield and berry size responses in 'Florida Radiance' were linear increases, whereas those in 'Florida127' were quadratic with peaks occurring at an N rate of 1.57 kg/ha/d. In 'Florida Radiance', SSC showed a negative linear response with a slope of –0.91 (0.91 °Brix decrease per 1 kg/ha/d N increase) during the early season. Therefore, optimization of early-season N fertilization is an important strategy to improve the profitability of winter strawberry production. Our previous results also suggest that N fertilization that considers cultivar-dependent dose-responses can improve fertilizer use efficiency while minimizing fruit quality loss and environmental pollution risks. Currently, many Florida growers apply N at high rates between 1.96 and 2.24 kg/ha/d during the early season, instead of applying pre-plant N, and then gradually switch to lower rates [14]. However, little is known about how high early-season N rates influence the sensory and physicochemical profiles of strawberries grown under subtropical climate conditions.

This study aimed to evaluate season and early-season N fertilization effects on yield and physicochemical attributes of 'Florida Radiance' strawberry grown under subtropical climate conditions. We used 'Florida Radiance' in this study because our previous work demonstrated its high sensitivity to N fertilization in terms of both yield and quality. At the time this study was conducted, 'Florida Radiance' was the leading strawberry cultivar grown in Florida [15]. This cultivar is grown widely in other winter strawberry production regions, including southwest Spain and Queensland, Australia [16].

## 2. Materials and Methods

### 2.1. Experiment Site

Strawberry field experiments were conducted at the University of Florida's Gulf Coast Research and Education Center in Balm, Florida, United States (latitude 27°76′, longitude 82°23′ W; elevation 39 m), during the 2015–2016 and 2016–2017 growing seasons. Plants were grown under open field conditions. The soil at this study site is classified as Myakka fine sand (sandy, siliceous, Hyperthermic Oxyaquic Alorthods), typical of the strawberry production area in west-central Florida. Soil samples were collected from 0 to 15 cm depth immediately before transplanting. In both seasons, soil pH was 6.8, and organic matter content was 0.8%. Nitrate- and ammonium-N contents were less than 2 mg/kg. In Florida, very low levels of inorganic N at pre-plant are common because of sandy soil and heavy rainfall during the rainy season (June to September), making residual soil N susceptible to leaching loss.

### 2.2. Plant Material

A major strawberry cultivar in Florida, 'Florida Radiance', was used in this study. Bare-root seedlings were shipped from a commercial nursery (G.W. Allen Nursery, Centreville, NS, Canada) to the study site and stored at 2 °C until transplanting.

### 2.3. Field Preparation and Crop Management

Growth conditions and management practices were similar to those used by local strawberry growers. In Florida, strawberry is grown as an annual crop using plasticul-

ture [17]. The raised beds used in this study were spaced 122 cm apart and measured 81 cm wide at the base, 71 cm wide at the shoulders, and 25 cm high at the bed top. During the bed preparation, beds were fumigated with PicClor 60 (TriEst Ag Group, Greenville, NC, USA) at 303 kg/ha to control weeds, nematodes, and soil-borne diseases. The fumigant was delivered 20 cm deep using a standard fumigation rig with three knives per bed. Immediately after fumigation, one line of drip tubing (0.95 L/h/emitter, 30.5-cm emitter spacing) was installed 2.5 cm deep in the middle of each bed, and beds were covered with black high-density polyethylene mulch (0.02-mm thick).

Transplanting was performed by hand on 13 October 2015 and 3 October 2016 in the 2015–2016 and 2016–2017 seasons, respectively. Seedlings were planted in double rows per bed spaced 30 cm apart with 38 cm between plants within each row. The planting density was 43,056 plants per hectare. Because of the high temperature during establishment and root desiccation and injury, sprinkler irrigation is required to promote plant survival until plants are fully established [18]. Therefore, sprinkler irrigation was run during daylight hours (9 h/d) for 13 to 14 days upon transplanting. Thereafter, plants were watered daily by drip irrigation. Standard pest management practices used by local commercial strawberry growers were followed [19].

*2.4. Fertilization Treatments*

Fertilization treatments tested in this study are described in Table 1. To avoid the risk of nitrate leaching, fertilization was performed after bare-root transplants were established in the field with sprinkler irrigation. In both growing seasons, plants were treated with three N rates (1.12, 1.68, and 2.24 kg/ha/d) over 21 days during the early vegetative growth stage. Thereafter, all plants were treated with the same N rate of 1.12 kg/ga/d until the end of the season. Total N inputs ranged from 153.6 to 177.1 kg/ha in the 2015–2016 season and 158.0 to 181.6 kg/ha in the 2016–2017 season. The fertilization schedule was determined based on the commercial grower practice used in Florida. The highest N rate used in this study is a typical N rate used by commercial strawberry growers in Florida.

**Table 1.** Nitrogen (N) fertilization treatments tested for 'Florida Radiance' strawberry in the 2015–2016 and 2016–2017 growing seasons.

| Season | Treatment # | Early Season N Rate [1] | | Mid–Late Season N Rate [2] | | Total N Input |
|---|---|---|---|---|---|---|
| | | (kg/ha/d) | (kg/ha) | (kg/ha/d) | (kg/ha) | (kg/ha) |
| 2015–2016 | 1 | 1.12 | 23.5 | 1.12 | 130.0 | 153.6 |
| | 2 | 1.68 | 35.3 | 1.12 | 130.0 | 165.3 |
| | 3 | 2.24 | 47.1 | 1.12 | 130.0 | 177.1 |
| 2016–2017 | 1 | 1.12 | 23.5 | 1.12 | 134.5 | 158.0 |
| | 2 | 1.68 | 35.3 | 1.12 | 134.5 | 169.8 |
| | 3 | 2.24 | 47.1 | 1.12 | 134.5 | 181.6 |

[1] From 26 October to 15 November 2015 (21 days) in the 2015–2016 season and from 17 October to 6 November 2016 (21 days) in the 2016–2017 season. [2] From 16 November 2015 to 10 March 2016 (150 days) in the 2015–2016 season and from 24 October 2016 to 6 March 2017 (155 days) in the 2016–2017 season.

Urea ammonium nitrate (32N–0P–0K) was used for N fertilization in both growing seasons. Other macro- and micro-nutrients were applied using a custom blend fertilizer that does not contain nitrogen (0N–0.9P–3.3K). Fertilization was performed through drip irrigation. Total phosphorous (P) and potassium (K) inputs were 23.0 and 84.5 kg/ha, respectively, in the 2015–2016 season, and 23.7 and 86.9 kg/ha, respectively, in the 2016–2017 season.

*2.5. Yield*

Strawberries were harvested twice a week from mid-November to early March in both seasons. November to December harvests were classified as the early-season yield, and January to March harvests were classified as the late-season yield. Fruit grading was performed according to the U.S. Department of Agriculture grading standards [20]. Marketable berries included both U.S. No. 1 and No. 2 grades with a minimum weight of

10 g. Other berries were graded as unmarketable. The number and fresh weight (FW) of marketable and unmarketable berries were recorded. The ratio of marketable yield (yield, kg/ha) and total fertilizer N input ($N_F$, kg/ha) was calculated as a measure of nitrogen use efficiency [21].

*2.6. Fruit Quality Analysis*

Marketable berries were sampled from peak harvest for fruit quality analysis. In the 2015–2016 season, 10 marketable berries were randomly selected per replication on 7 March 2016. In the 2016–2017 season, 9 to 14 marketable berries were randomly selected per replication on 15 February 2017.

Instrumental color analysis was conducted by taking two-color measurements on opposite sides of the fruit in the equatorial region. A hand-held tristimulus reflectance colorimeter (Model CR-400, Minolta Co., Ltd., Osaka, Japan) equipped with a glass light-protection tube (CR-A33, Minolta Co., Ltd., Osaka, Japan) was used. The color was recorded using the CIE-L*a*b* uniform color space (CIE-Lab), where L* indicates lightness (0 = black and 100 = white), a* indicates chromaticity on a green (–) to red (+) axis, and b* chromaticity on a blue (–) to yellow (+) axis. Numerical values of a* and b* were converted into hue angle and chroma using the Konica-Minolta CR-400 Utility Sofware (©2011–2019 Konica Minolta, Inc., Tokyo, Japan).

The firmness of each strawberry was measured using a TA. XT Plus Texture Analyzer (Texture Technologies Corp., Hamilton, MA, USA) as described by Nunes et al. [22]. The force required to compress the fruit by 3 mm was recorded in kgf and converted to Newton (N = kgf × 9.8).

Three replicated samples of 10 individual fruit per treatment were homogenized using a laboratory blender at high speed for 2 min. The resulting puree was immediately frozen and kept at –30 °C until used. Titratable acidity (TA) and SSC were determined according to the procedure described by Nunes et al. [23].

Total phenolic compounds were measured using the Folin–Ciocalteu reagent as described by Nunes et al. [24]. Anthocyanins were extracted in 0.5% (*v/v*) HCl in methanol and measured using the procedure described by Nunes et al. [24].

*2.7. Weather*

Weather data at the study site were obtained from the Florida Automated Weather Network (https://fawn.ifas.ufl.edu/data/reports/ accessed on 28 April 2019). Table 2 shows the weather data during the entire season (from planting to final harvest) and the fruit development period (from anthesis to fruit ripening) for the berries sampled for fruit quality analysis. Average temperature, relative humidity, and solar radiation values were calculated using daily average values. Cumulative rainfall data were calculated using daily rainfall values.

**Table 2.** Weather conditions in the 2015–2016 and 2016–2017 growing seasons [1].

| Season | Duration | Temperature (°C) | | | RH | SR | Rainfall |
|---|---|---|---|---|---|---|---|
| | | **Max** | **Min** | **AVR** | **(%)** | **(W/m²)** | **(mm)** |
| 2015–2016 | Entire season [2] | 25.2 | 13.8 | 19.0 | 79.7 | 148 | 325 |
| | Fruit development [3] | 23.0 | 8.9 | 15.7 | 72.0 | 197 | 32 |
| 2016–2017 | Entire season [4] | 26.2 | 13.6 | 19.4 | 78.4 | 151 | 95 |
| | Fruit development [5] | 24.3 | 11.2 | 17.5 | 77.8 | 156 | 40 |

Max = maximum. Min = minimum. AVR = average. RH = relative humidity. SR = solar radiation. [1] Temperature, RH, and SR data are the averages calculated using daily average values. Rainfall data are the cumulative values. Source: Florida Automated Weather Network (https://fawn.ifas.ufl.edu/data/reports/) (accessed on 28 April 2019). [2] From 13 October 2015 to 10 March 2016 (150 days). [3] Fruit development period for sampled berries (from anthesis to fruit ripening): from 8 February to 6 March 2016 (28 days). [4] From 13 October 2015 to 10 March 2016 (155 days). [5] Fruit development period for sampled berries (from anthesis to fruit ripening): from 18 January to 14 February 2016 (28 days).

*2.8. Experimental Design and Statistical Analysis*

In both growing seasons, N fertilization treatments were arranged in a randomized complete block design. Each treatment had four plots (replicates), and each plot consisted of 20 plants.

All data were analyzed by the generalized linear mixed model procedure (PROC GLIMMIX) in the SAS statistical software (SAS 9.4; SAS Institute Inc., Cary, NC, USA). Season, early-season N rate, and season × early season N rate interaction were considered fixed factors, whereas block and block × season interaction were considered random factors.

The best model was selected based on the smallest corrected Akaike information criterion. Marketable yield, average berry weight, texture, anthocyanin, and phenolics data were modeled with the lognormal distribution (DIST = LOGNORMAL). For model parameter estimation, boundary constraints on covariance were removed (NOBOUND), and degrees of freedom for the fixed effects were adjusted by the Kenward–Roger degrees of freedom approximation (DDFM = KR). Other data were modeled with the beta distribution (DIST = BETA), and model parameters were estimated by maximum likelihood estimation with quadrature approximation (METHOD = QUAD) and default bias-corrected sandwich estimators (EMPIRICAL = MBN).

Log-transformed data were back-transformed by exponentiating the sum of the least square mean and the correction factor. Comparisons of least square means were performed using the Tukey–Kramer test. For all data analyzed, $p$ values equal to or less than 0.05 were considered statistically significant. Back-transformed or rescaled data are reported in this study.

Linear regression was performed using SigmaPlot (Systat Software Inc., San Jose, CA, USA) to assess the linear correlation between two measurement variables. The correlation was considered significant when the slope was significantly different from zero ($p \leq 0.05$).

## 3. Results

*3.1. Marketable Yields*

Marketable yield data were pooled by each main effect because the season × early season N rate interaction was not significant (Figure 1). Marketable yield was not affected by seasons, but it was significantly different among treatments with varying early season N rates. Increasing the early-season N rate increased late- and total-season yields by 19% to 21% (14.86 vs. 17.62–17.93 t/ha) and 15% to 18% (18.76 vs. 21.63–22.17 t/ha), respectively. The yields at high N rates were comparable to local commercial yields of the same seasons.

*3.2. Average Berry Weight*

Average berry FW data were pooled by each main effect because the season × early season N rate interaction was not significant (Figure 2). Although the average FW of berries harvested during the early season was similar in both seasons (15.95–16.08 g), that during the late-season was 20% greater in the 2015–2016 season than in the 2016–2017 season. As a result, the total-season average berry FW was 14% greater in the 2015–2016 season than in the 2016–2017 season. Early season N rate had no significant effect on the average berry FW.

*3.3. Linear Correlations between Berry Size and Yield or Berry Number and Yield*

Marketable yield had a strong positive correlation with berry number ($r^2 = 0.698$), but it had no significant correlation with the average berry FW (Figure 3).

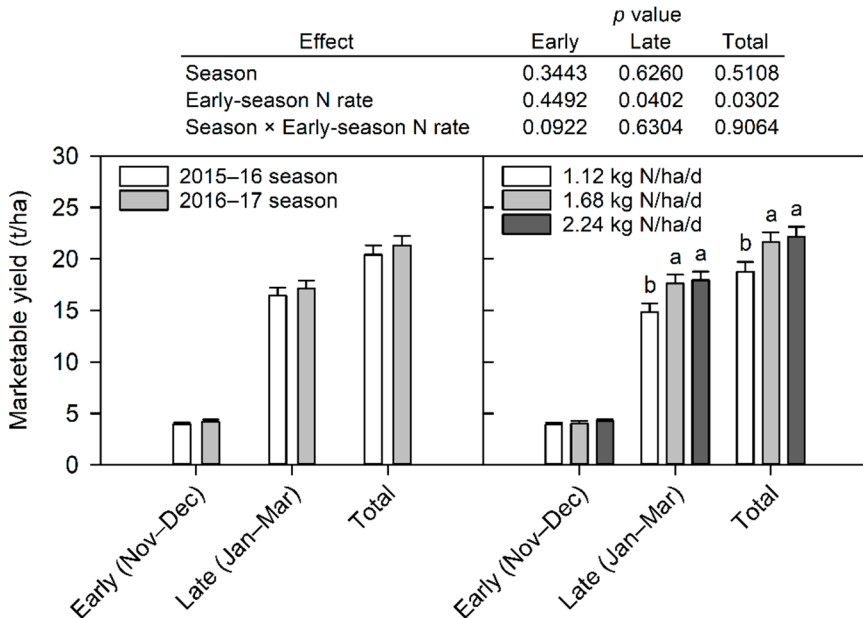

**Figure 1.** Marketable yield of 'Florida Radiance' strawberry as affected by seasons and early-season nitrogen (N) fertilization rates. Fertilization treatments are as described in Table 1. November to December and January to March harvests were classified as early and late season yields, respectively. Data are means ± SE. Data were pooled by each main effect, as they were not significantly affected by the season × early season N rate interaction ($p > 0.05$). Within each yield group, bars followed with the same letter are not significantly different (Tukey–Kramer test, $p \leq 0.05$).

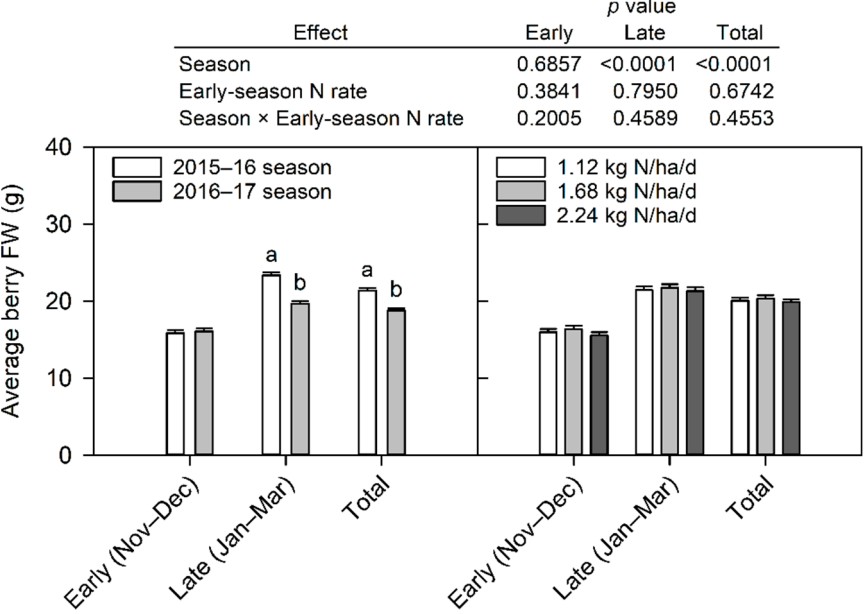

**Figure 2.** Average berry fresh weight (FW) of 'Florida Radiance' strawberry as affected by season and early-season nitrogen (N) fertilization rate. Fertilization treatments are as described in Table 1. November to December and January to March harvests were classified as early and late season yields, respectively. Data are means ± SE. Data were pooled by each main effect, as they were not significantly affected by the season × early season N rate interaction ($p > 0.05$). Within each yield group, bars followed with the same letter are not significantly different (Tukey–Kramer test, $p \leq 0.05$).

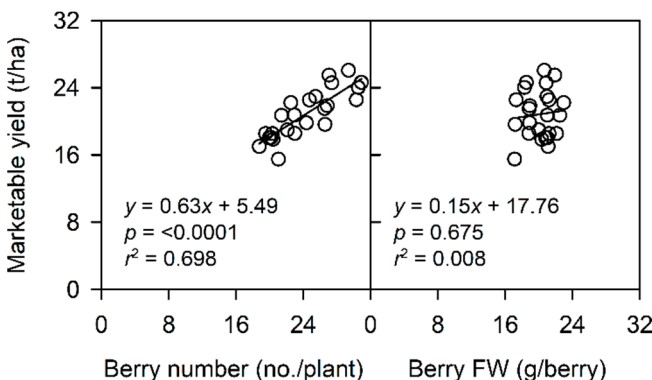

**Figure 3.** Linear regression between berry number and yield or average berry fresh weight (FW) and yield of 'Florida Radiance' strawberry. Data from the 2015–2016 and 2016–2017 seasons are included in the figure. The correlation was considered significant when the slope was significantly different from zero ($p \leq 0.05$).

### 3.4. Ratio of Yield and Total Fertilizer N Input

The highest yield/$N_F$ ratio was recorded at the early-season N rate of 2.24 kg/ha/d (129.0), followed by 1.68 kg/ha/d (123.6) and then 1.12 kg/ha/d (120.4). However, neither season, early-season N rate, nor season × early-season N rate interaction significantly affected yield/$N_F$ ratio (Figure 4).

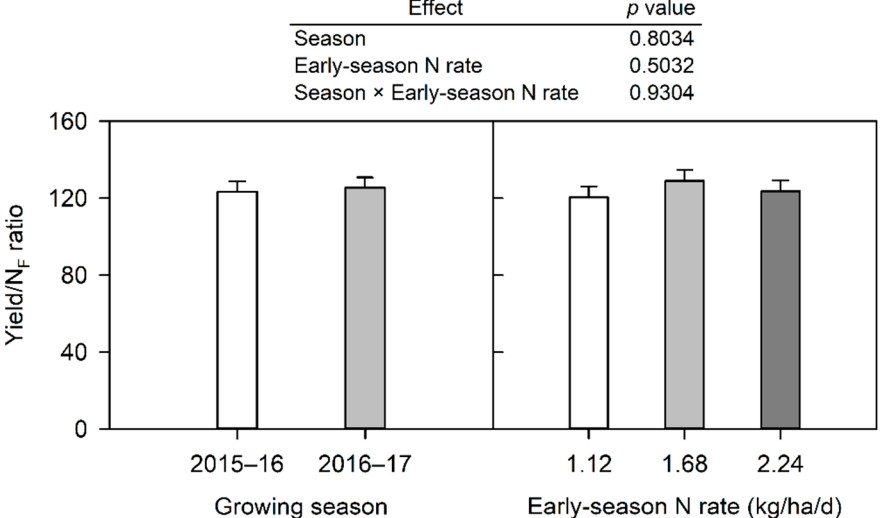

**Figure 4.** The ratio of marketable yield (yield, kg/ha) and total fertilizer nitrogen (N) input ($N_F$, kg/ha) of 'Florida Radiance' strawberry as affected by season and early-season N fertilization rate. Fertilization treatments are as described in Table 1. Data are means ± SE. Data were pooled by each main effect, as they were not significantly affected by the season × early season N rate interaction ($p > 0.05$).

### 3.5. Fruit Color and Texture

Color space and appearance parameters measured to characterize fruit color were significantly different between the two seasons, except for the lightness value, L* (Table 3). Overall, the fruit color was redder but less vivid (lower hue and chroma values) in the 2015–2016 season than in the 2016–2017 season. None of the color parameters was significantly affected by the early-season N rate. However, there was a tendency for fruit color to be lighter (higher L*), more vivid (higher chroma), and less red (higher hue) at the highest early-season N rate than at lower rates.

**Table 3.** Color of 'Florida Radiance' strawberry as affected by season and early-season nitrogen (N) fertilization rate [1].

| Season | Early-Season N Rate [2] | Color | | | | | | | | | Texture [3] |
|---|---|---|---|---|---|---|---|---|---|---|---|
| | (kg/ha/d) | L* | a* | | b* | | Chroma | | Hue | | (N) |
| 2015–2016 | | 38.0 | 24.1 | b | 7.4 | b | 25.4 | b | 16.4 | b | 4.40 |
| 2016–2017 | | 38.9 | 27.5 | a | 11.1 | a | 29.8 | a | 21.6 | a | 4.42 |
| | 1.12 | 38.1 | 26.0 | | 9.3 | | 27.7 | | 18.9 | | 4.49 |
| | 1.68 | 38.5 | 25.6 | | 9.0 | | 27.3 | | 18.8 | | 4.32 |
| | 2.24 | 39.0 | 26.7 | | 10.5 | | 28.8 | | 20.6 | | 4.43 |
| | | *p* value | | | | | | | | | |
| Season | | 0.0847 | 0.0005 | | <0.0001 | | 0.0002 | | <0.0001 | | 0.9719 |
| Early-season N rate | | 0.5722 | 0.9257 | | 0.7132 | | 0.8711 | | 0.5725 | | 0.7898 |
| Season × Early-season N rate | | 0.5718 | 0.5668 | | 0.2290 | | 0.4910 | | 0.0887 | | 0.0217 |

[1] Means in a column with the same letter or no letter are not significantly different (Tukey–Kramer test, $p \leq 0.05$). [2] Fertilization treatments are as described in Table 1. [3] Texture at early-season N rates of 1.12, 1.68, and 2.24 kg/ha/d was 4.03, 4.57, and 4.63 N in the 2015–2016 season, respectively, and 4.84, 4.08, and 4.33 N in the 2016–2017 season, respectively. Despite the significant season × early season N rate interaction, no significant difference was found among the treatment means (Tukey–Kramer test, $p \leq 0.05$).

Fruit texture was not significantly affected by season and early-season N rate (Table 3). Although the season × early season N rate interaction was significant, treatment means showed no significant difference according to the Tukey–Kramer test.

### 3.6. Chemical and Sensory Attributes

Chemical and sensory attributes showed both positive and negative changes between the two seasons (Table 4). Strawberries harvested in the 2015–2016 season had lower pH, TA, and SSC but higher SSC/TA ratio and anthocyanin content than those in the 2016–2017 season. The most notable change occurred in the anthocyanin levels, which showed a 47% higher value in the 2015–2016 season than in the 2016–2017 season. Early-season N rate had no significant effect on any chemical characteristics measured in this study. There was a significant negative correlation between hue and anthocyanin content ($r^2 = 0.575$) (Figure 5).

**Table 4.** Chemical characteristics of 'Florida Radiance' strawberry as affected by season and early-season nitrogen (N) fertilization rate [1].

| Season | Early-Season N Rate [2] | pH | | TA | | SSC | | SSC/TA | | Anthocyanins | | Phenolics |
|---|---|---|---|---|---|---|---|---|---|---|---|---|
| | (kg/ha/d) | | | (%) | | (%) | | | | (mg/100 g FW) | | (mg/100 g FW) |
| 2015–2016 | | 3.51 | b | 0.649 | b | 5.69 | b | 8.78 | a | 17.2 | a | 180 |
| 2016–2017 | | 3.58 | a | 0.864 | a | 6.32 | a | 7.36 | b | 11.7 | b | NA |
| | 1.12 | 3.54 | | 0.768 | | 6.08 | | 8.01 | | 14.7 | | 185 |
| | 1.68 | 3.54 | | 0.758 | | 5.87 | | 7.88 | | 13.9 | | 175 |
| | 2.24 | 3.59 | | 0.795 | | 6.27 | | 8.04 | | 13.4 | | 184 |
| | | *p* value | | | | | | | | | | |
| Season | | 0.0238 | | <0.0001 | | <0.0001 | | <0.0001 | | <0.0001 | | NA |
| Early-season N rate | | 0.7292 | | 0.9963 | | 0.1345 | | 0.4706 | | 0.1704 | | 0.3955 |
| Season × Early-season N rate | | 0.0579 | | 0.5703 | | 0.7763 | | 0.7398 | | 0.2393 | | NA |

[1] Means in a column with the same letter or no letter are not significantly different (Tukey–Kramer test, $p \leq 0.05$). [2] Fertilization treatments are as described in Table 1. TA = titratable acidity. SSC = soluble solids concentration. FW = fresh weight. NA = data not available.

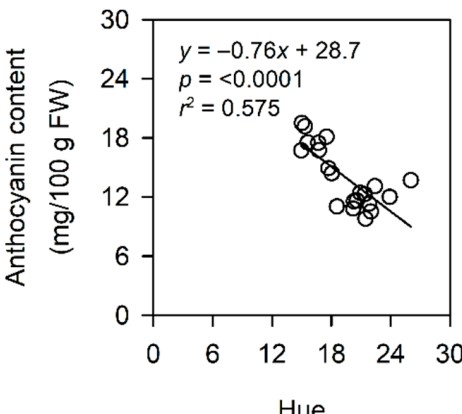

**Figure 5.** Linear regression between berry hue and anthocyanin content of 'Florida Radiance' strawberry. Data from the 2015–2016 and 2016–2017 seasons are included in the figure. The correlation was considered significant when the slope was significantly different from zero ($p \leq 0.05$).

## 4. Discussion

### 4.1. Yield Increases by High Early-Season N Fertilization Are Long-Term and Consistent

In winter strawberry production, early-season N fertilization plays an important role in improving canopy growth and yields, especially when pre-plant N is not applied [13]. For example, in Florida, where the majority of winter strawberries are produced in the United States, many strawberry growers apply N at high rates (1.96–2.24 kg/ha/d) during the early season, instead of applying preplant N, and then gradually switch to lower rates [14]. Using model fitting analysis, our previous study demonstrated that leaf area and marketable yield of 'Florida Radiance' increased linearly with increasing the early-season N rate from 0.22 to 2.02 kg/ha/d [13]. In this study, although N rate was increased only over 21 days during the vegetative growth stage, yield increases by high early-season N rates (1.68 and 2.24 kg/ha/d) became greater and significant later in the season, indicating the long-term effect of early-season N fertilization on crop productivity. Furthermore, the strong positive correlation between berry number and marketable yield confirms the importance of the early establishment of a productive canopy for increasing fruit set in winter strawberry production.

It is important to note that yield increases with increasing the early-season N rates were consistent over the two seasons, despite the differences in weather conditions, especially rainfall. In this study, small amounts of N were applied daily through drip irrigation. Therefore, the consistent results are likely due to the frequent delivery of N to the root zone that often improves fertilizer N use efficiency [25].

The high responsiveness to early-season N fertilization is an interesting observation, considering the relatively low nutrient demand by young plants during the treatment period. This observation may be explained by the limited nutrient uptake capacity of bare-root transplants during establishment. Bare-root transplants are the primary transplant type used for commercial strawberry production in Florida [18]. Bare-root transplants are shipped in boxes from nurseries in temperate regions, primarily Canada and northern California, United States, to strawberry production farms in Florida after cleaning the soil from the roots. As a result, bare-root transplants do not have many active roots at the time of transplanting, so their nutrient uptake capacity is limited until sufficient new roots are developed. Therefore, during establishment, high concentrations of fertilizer solution applied to the root zone appear to improve the nutrient uptake by the transplants.

### 4.2. Optimum Early-Season N Fertilization Can Increase Yield with a Minimal Risk of Increasing N Loss and Environmental Pollution Risks

It is important to consider not only yields but also fertilizer N use efficiency when optimizing nutrient management. Excessive N fertilization can increase the potential of N

loss through ammonia volatilization or leaching and runoff of nitrate [26]. In particular, the shallow root system of strawberry plants [27] and Florida's sandy soil increase the risk of nitrate leaching [26]. Nitrate is a common contaminant of groundwater, and nitrate levels above 10 mg/L in drinking water pose a health risk, especially for infants and pregnant women [28]. Excess nitrate also leads to water quality deterioration and nuisance algal growth in surface waters and groundwater-fed systems [26,28]. In this study, increasing the early-season N rate from 1.12 to 1.68 and 2.24 kg/ha/d increased marketable yield by 15% and 18%, respectively, at the expense of increasing the total N input by 11.8 and 23.5 kg, respectively. As a result, the highest yield/$N_F$ ratio was recorded at the early-season N rate of 2.24 kg/ha/d, followed by 1.68 kg/ha/d and then 1.12 kg/ha/d. These results suggest that applying N at a higher rate (1.12 vs. 1.68 kg/ha/d) during the early season is an effective production strategy to increase yield with a minimal risk of increasing N loss and consequent environmental pollution risks.

### 4.3. Season Has Greater Effects on Fruit Quality than Early-Season N Fertilization

Fruit quality showed remarkable seasonal variations, but none of the fruit quality attributes showed a significant response to the early-season N rate. Similar findings have been reported in diverse strawberry producing regions [5,7,11,29,30]. For example, in 'Primoris' strawberry grown in Spain, Cardeñosa et al. [7] found only minor differences in color, pH, TA, SSC, and ascorbic acid contents among different N treatments. By contrast, ascorbic acid, total phenolics, and total anthocyanin contents were higher when strawberries were harvested in March than in January. In four strawberry cultivars grown in Argentina, Agüero et al. [5] also found that variations in nutrient supply did not affect fruit quality, but firmness, color, TA, and SCC depended on air temperature and rainfall. Kelly et al. [2] reported that differences in the physicochemical characteristics of Florida strawberry cultivars between seasons and years were strongly attributed to climate conditions during the growing season and the ripening stage at harvest to some extent. In general, strawberries harvested at the full red stage tend to have higher SSC, anthocyanin, and sugar contents than those at the three quarter-red stage [31]. Higher temperatures and prolonged exposure to sunlight are known to boost fruit metabolism, prompting the phenylpropanoid pathway particularly to synthesize anthocyanin compounds [32]. Our results and findings in these previous studies suggest that season has greater effects on fruit quality than early-season N fertilization. The lack of significant effects of early-season N rate on fruit quality also suggests that optimization of N fertilization can increase strawberry yields with a minimal impact on source-sink relationships.

### 4.4. Impacts of Environmental Conditions on Physicochemical Profiles

It is well-documented that environmental conditions can affect various physicochemical attributes of strawberries, including fruit color, SSC, TA, ascorbic acid, phenolics, and anthocyanins [5,7,8,29,33]. Our results showed both positive and negative season effects on fruit quality, depending on quality attributes measured. Environmental conditions also differed significantly between the two seasons, indicating an interaction between quality attributes and environmental conditions. Similar results have been reported in previous studies. For example, in four strawberry cultivars grown under subtropical climate conditions in Argentina, Agüero et al. [5] found that increases in air temperature later in the season contributed to increases in redness and TA but decreases in fruit weight and firmness. In a study conducted with 'Primoris' strawberry grown in Spain, Cardeñosa et al. [7] reported that decreased TA and increased SSC in the late season were due partly to increased light intensity.

Sugars are the major soluble solids in fruit juice, and SSC is a robust analytical measurement with a strong correlation with total sugars in strawberries [34]. By contrast, TA is a good indicator for sourness [34]. It is reported that SSC, SSC/TA, and TA are all important sensory attributes that influence the perception of sweetness and flavor [34]. In plants, simple carbohydrates formed by photosynthesis are used as building blocks for various

carbohydrates and amino acids [12]. Under optimum growing conditions, increasing light intensity and temperature promotes photosynthesis [12,35]. In this study, lower SSC and TA were recorded when radiation was 26% higher, and maximum, minimum, and average temperatures were 6%, 21%, and 10% lower, respectively, compared to the other season. Therefore, it can be speculated that the negative effect of lower temperatures on photosynthesis may have been greater than the positive effect of higher solar radiation, resulting in lower SSC and TA.

Anthocyanin synthesis is triggered by ultraviolet radiation, and increased anthocyanin accumulation is often attributed to exposure to high-intensity light during fruit development [7,33,36]. Anttonen et al. [36] reported that anthocyanin content was up to 16% higher in 'Bounty' strawberries grown without shading than those with 32% shading. Similarly, Cervantes et al. [33] found that canopy shading significantly reduced anthocyanin content in four strawberry cultivars grown in Spain. In this study, anthocyanin content was 47% higher in the season with 26% higher solar radiation and 6% to 21% lower temperature, RH, and rainfall during the fruit development period, confirming the critical role of solar radiation in anthocyanin accumulation in strawberries.

Anthocyanins contribute to the red coloration of strawberries and act as a protective barrier from the damaging effects of ultraviolet radiation [32]. The significant negative correlation between hue angle and anthocyanin content obtained in this study also suggests that anthocyanin content increases as fruit color become redder in strawberries. Therefore, the redness of strawberries appears to be a good indicator of anthocyanin levels, which are known to have high antioxidant properties and beneficial health effects [37].

## 5. Conclusions

In this study, increasing the early-season N rate increased marketable yield by up to 18% with no negative effects on physicochemical characteristics of 'Florida Radiance' strawberry under subtropical climate conditions. These results suggest that the use of high N rates during the early season is an effective strategy to improve the profitability of winter strawberry production. Importantly, this unique fertilization technique eliminates the need for pre-plant N application, which can be subjected to nitrate leaching or runoff by sprinkler irrigation during establishment. Furthermore, this fertilization technique has a minimal risk of compromising N fertilizer use efficiency, as it increases N rate only during the early vegetative growth stage. For 'Florida Radiance' grown under subtropical climate conditions, the optimum early-season N rate appears to be 1.68 kg/ha in terms of yield and N use efficiency responses.

This study also demonstrates significant seasonal changes in various fruit quality attributes, including fruit color, SSC, TA, SSC/TA, and anthocyanin content. Depending on quality attributes, both positive and negative changes were observed between the two seasons, during which environmental conditions differed significantly. To accurately assess season effects on overall fruit quality, it is important to record various weather variables, including temperature, RH, rainfall, light intensity, and cloudiness.

**Author Contributions:** Conceptualization, S.A.; methodology, S.A. and M.C.d.N.N.; statistical analysis, S.A.; investigation, S.A.; resources, S.A. and M.C.d.N.N.; data curation, S.A. and M.C.d.N.N.; writing—original draft preparation, S.A.; writing—review and editing, S.A. and M.C.d.N.N.; visualization, S.A.; supervision, S.A. and M.C.d.N.N.; project administration, S.A.; funding acquisition, S.A. All authors have read and agreed to the published version of the manuscript.

**Funding:** This research was funded by the Florida Strawberry Research and Education Foundation.

**Institutional Review Board Statement:** Not applicable.

**Informed Consent Statement:** Not applicable.

**Data Availability Statement:** The data presented in this study are available on request from the corresponding author.

**Acknowledgments:** We thank all members of Horticultural Crop Physiology Lab at the Gulf Coast Research and Education Center for their technical assistance.

**Conflicts of Interest:** The authors declare no conflict of interest.

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
