# Peer review of "Season and Nitrogen Fertilization Effects on Yield and Physicochemical Attributes of Strawberry under Subtropical Climate Conditions"

_agronomy, doi:10.3390/agronomy11071391_

Round 1

Reviewer 1 Report

The study is interesting and has a great application however; I have few quires which should be addressed before it is considered further

Title can be changed, it is too long. Title should be crisp and more specific.

Please mention the age of cutting during the time of replanting

Authors mentioned seedlings were shipped from Nova Scotia to the study site, however, the genotype sued is Florida Radiance is this genotype also common in Nova Scotia region why was not it purchased from the local region from a local grower as authors mentioned that this genotype is common in Florida regions. Could have been more authenticated if used locally

Whether experiments were performed in an open field or closed conditions please mention it.

Please also mention how was the N rates selected, is based on the data provided by local growers of strawberries. Also, what was the control taken? It seems no positive and negative control was taken (without N)

Photographs of berries changing its color under N supply could have been also added to correlate with your statistical data presented.

Author Response

Thank you so much for your constructive comments and suggestions! We truly appreciate your time and the opportunity to revise the paper!

Below we have listed my individual responses to your comments, describing how each of your concern and comment was addressed in the revision.

  1. Title can be changed, it is too long. Title should be crisp and more specific.

We shortened the title. New title is: Season and Nitrogen Fertilization Effects on Yield and Physicochemical Attributes of Strawberry under Subtropical Climate Conditions

  1. Please mention the age of cutting during the time of replanting

Thank you for the suggestion. The age of seedlings would be useful information for plug transplants, which are propagated using immature runners separated from mother plants at a specific growth stage. In this study, however, we used bare-root transplants, which are basically mature runners grown in an open field without being removed from mother plants until the harvesting time. It is difficult to measure the age of bare-root transplants. Therefore, we decided not to include this information in this manuscript.

  1. Authors mentioned seedlings were shipped from Nova Scotia to the study site, however, the genotype sued is Florida Radiance is this genotype also common in Nova Scotia region why was not it purchased from the local region from a local grower as authors mentioned that this genotype is common in Florida regions. Could have been more authenticated if used locally

If seedlings are available locally, we agree that it would be better to obtain the seedlings from local nurseries. However, strawberry seedlings are not available in Florida because our climate is not suitable for strawberry transplant production. All strawberry growers in Florida purchase transplants from nurseries in temperate regions, primarily Canada and northern California. Please see L313-317.

  1. Whether experiments were performed in an open field or closed conditions please mention it.

Changes were made as suggested (80-81). Thank you for the suggestion!

  1. Please also mention how was the N rates selected, is based on the data provided by local growers of strawberries. Also, what was the control taken? It seems no positive and negative control was taken (without N)

We added new sentences to explain how we selected N rates in this study (L120-122). Our previous study tested a wider N rate range and analyzed the rate effect using model fitting analysis. In this study, we tested the effect of increasing N using a relatively narrow range that provides more practical information.

  1. Photographs of berries changing its color under N supply could have been also added to correlate with your statistical data presented.

Thank you so much for the suggestion! We agree that photographs would be good supplemental information. We will definitely consider taking photos in our future study.

Reviewer 2 Report

The article is overall very interesting and what I like the most, it is practical and even a strawberry producer can find useful information. What I missed and also had a problem to understand is that I don’t know the production technology – as the production technology differs a lot in different countries. It is unique in Denmark, totally different in Austria and specific in subtropical region. I think that some general information, just a short explanation of what kind of production technology is used in Florida. Since this article will be read also in other countries, this information would be in great use for better understanding.

The main concern that can bring droughts is that I couldn’t find the information about the available nitrogen in soil/plant before and after the fertilization. What is the recommended dose of nitrogen? How many nitrogen can be applied in your country, do you have any limits? It is also important to know if the results effect only fertilization with N or also the deficiencies/excesses of other nutrients. It is important to know if the plants were in stress because of the deficit of excess of the nitrogen, especially when we are then looking closely to the content of secondary metabolites.

My other concern is that the analyses that were performed for sensory evaluation of strawberries are very simple and too general.  In this case I did expect at least some HPLC phenolic analyses to look closely to the inner quality based on the individual phenolic. Also for primary metabolites, such as individual sugars, organic acids, vitamin C. Anthocyanins – the colour of the fruits is hardly affected because of the nitrogen fertilization, hydroxibenzoic acids, flavanols and hydroxycinnamic acids are usually affected. 

I didn’t find the information about the number of fruits, have you monitored it? It is important when we talk about yield to know if we have many of small fruits (10g as you sad) or less/more bigger fruits.

And one more suggestion, rearrange the abstract, I think it doesn’t provide the necessary information about the experiment and which measurements were performed. I couldn’t find what unit kg/ha/d means? And what it means kg/ga/d?

References (21, 28, 29,…) use the abbreviations of the journal,…

Author Response

We greatly appreciate your time to review our manuscript. All comments are extremely helpful to improve the manuscript. Thank you so much!

Below we have listed my individual responses to your comments, describing how each of your concern and comment was addressed in the revision.

  1. The article is overall very interesting and what I like the most, it is practical and even a strawberry producer can find useful information. What I missed and also had a problem to understand is that I don’t know the production technology – as the production technology differs a lot in different countries. It is unique in Denmark, totally different in Austria and specific in subtropical region. I think that some general information, just a short explanation of what kind of production technology is used in Florida. Since this article will be read also in other countries, this information would be in great use for better understanding.

Thank you so much for the positive feedback and for constructive suggestions! We revised the method section as suggested (L80-88; L94-96; L107-109; L120-122) and added two references (17 and 19) that describe strawberry production practices used in Florida.

  1. The main concern that can bring droughts is that I couldn’t find the information about the available nitrogen in soil/plant before and after the fertilization. What is the recommended dose of nitrogen? How many nitrogen can be applied in your country, do you have any limits? It is also important to know if the results effect only fertilization with N or also the deficiencies/excesses of other nutrients. It is important to know if the plants were in stress because of the deficit of excess of the nitrogen, especially when we are then looking closely to the content of secondary metabolites.

Thank you so much for your suggestion! We added initial inorganic N information in L83-89, and commercial N fertilization information in L120-122.

  1. My other concern is that the analyses that were performed for sensory evaluation of strawberries are very simple and too general. In this case I did expect at least some HPLC phenolic analyses to look closely to the inner quality based on the individual phenolic. Also for primary metabolites, such as individual sugars, organic acids, vitamin C. Anthocyanins – the colour of the fruits is hardly affected because of the nitrogen fertilization, hydroxibenzoic acids, flavanols and hydroxycinnamic acids are usually affected.

We agree that the chemical analysis performed are basic (analytical color and texture, total acidity, SSC, total phenolic, and anthocyanin contents), but we believe they constitute a baseline and thus can be a good indicator of the effect of N fertilization on the overall quality of strawberries. We also agree that polyphenol profiles, namely individual polyphenols, can be affected by N fertilization. Still, overall, previous research indicates (although sometimes results are controversial) that weather conditions have a more significant impact on strawberry polyphenols than N fertilization. Although our laboratory is equipped with HPLCs, and we could have had performed HPLC analysis of individual sugars and polyphenols, these types of analysis are costly. Thus, budget limitations led us to decide on choosing basic chemical analysis. We appreciate your suggestion, and in our further research, we will take into consideration your suggestions.

  1. I didn’t find the information about the number of fruits, have you monitored it? It is important when we talk about yield to know if we have many of small fruits (10g as you sad) or less/more bigger fruits.

Yes, we collected both fruit number and weight data. Fruit number data are included in Figure 3. Because fruit number showed a very similar trend as yield as measured in t/ha, we decided to present the correlation between fruit number and yield. In this study, fruit number appeared to have a greater impact than fruit size, as shown in the correlation plots (Figure 3).

  1. And one more suggestion, rearrange the abstract, I think it doesn’t provide the necessary information about the experiment and which measurements were performed. I couldn’t find what unit kg/ha/d means? And what it means kg/ga/d?

kg/ga/d was a typo, and it was corrected in the revised manuscript. Thank you so much for the catch! Kg/ha/d means kg per hectare per day. Some information related to experiment methods were omitted because of the word limit, but we believe that most relevant information of this study are included in the abstract.

  1. References (21, 28, 29,…) use the abbreviations of the journal,…

Thank you for the catch! Corrections were made in the revised manuscript.

Reviewer 3 Report

A well written manuscript of sound science.  Well done.

Author Response

Thank you so much for reviewing the manuscript and for your positive feedback!

Below I have listed my individual responses to your comments, describing how each of your concern and comment was addressed in the revision.

  1. A well written manuscript of sound science. Well done.

Thank you so much for your comment!

Round 2

Reviewer 2 Report

Thank you for all the clarifications you have made in the manuscript and also for the new knowledge I have received about your production technology. As I am a soft fruit researcher this kind of information are very interesting to me. 

I think that the manuscript is well improved.